# Impact of Unexpected In-House Major COVID-19 Outbreaks on Depressive Symptoms among Healthcare Workers: A Retrospective Multi-Institutional Study

**DOI:** 10.3390/ijerph20064718

**Published:** 2023-03-07

**Authors:** Hideki Sato, Masaharu Maeda, Yui Takebayashi, Noriko Setou, Jiro Shimada, Yumiko Kanari

**Affiliations:** 1Department of Disaster Psychiatry, Fukushima Medical University School of Medicine, Fukushima 960-1295, Japan; 2Hospital Futaba Emergency General Medical Support Center, Fukushima Medical University, Fukushima 960-1295, Japan; 3Department of Emergency and Critical Care Medicine, Fukushima Medical University School of Medicine, Fukushima 960-1295, Japan; 4Fukushima Prefectural Government, Department of Health and Welfare, Fukushima Prefectural Headquarters for Coronavirus Infection Control, Fukushima 960-8670, Japan; 5Department of Public Health, Fukushima Medical University School of Medicine, Fukushima 960-1295, Japan

**Keywords:** COVID-19, major outbreak, healthcare worker, depressive symptom, suicidal ideation

## Abstract

The coronavirus disease 2019 (COVID-19) has significantly affected the mental health of healthcare workers (HCWs). The authors have provided psychosocial support to HCWs working in typical hospitals and nursing homes in Fukushima Prefecture, Japan, where major COVID-19 outbreaks have occurred since December 2020. This study retrospectively examines depressive symptoms before psychosocial interventions among HCWs working at typical hospitals and nursing homes experiencing in-house major COVID-19 outbreaks. We have offered psychosocial support in eight hospitals and nursing homes, obtaining data on the mental health status of 558 HCWs using the Patient Health Questionnaire-9. The study’s results indicate that 29.4% of HCWs have exhibited moderate or higher depressive symptoms, and 10.2% had suicidal ideation. Multiple logistic regression analysis showed that being a nurse was associated with higher depressive symptoms and suicidal ideation compared to other HCWs. In addition, multiple logistic regression analysis of Polymerase Chain Reaction-positive HCWs showed that being a nurse and the number of COVID-19-related symptoms was associated with high depressive symptoms. These results suggest that HCWs in typical hospitals and nursing homes experiencing major COVID-19 outbreaks are more likely to exhibit severe depressive symptoms, which may worsen if infected with COVID-19. This study’s findings expand the current understanding of HCWs’ depressive symptoms and the importance of psychosocial support during unexpected major outbreaks in healthcare facilities.

## 1. Introduction

The novel coronavirus (SARS-CoV-2) was first identified in Wuhan, Hubei Province, in December 2019 [1]. The coronavirus disease 2019 (COVID-19) spread rapidly worldwide, and the World Health Organization (WHO) declared a public health emergency of international concern on 30 January 2020. It characterized COVID-19 as a pandemic on 12 March. In Japan, COVID-19 was first identified in January 2020 and then spread nationwide [2]. As of 20 January 2023, the number of people infected with COVID-19 was approximately 664 million worldwide and 32 million in Japan [3].

Previous studies have reported that the COVID-19 pandemic has worsened the mental health of healthcare workers (HCWs) [4,5,6,7,8]. HCWs, such as nurses and nursing home staff, were essential to the COVID-19 public health response. However, HCWs were at a very high risk of infection as they were required to provide care in close proximity to patients [9]. Furthermore, they were burdened with work during the pandemic [10,11]. Major depressive symptoms in HCWs have been associated with suicidal ideation and the ‘moral injury’ of failing to provide adequate care, which violated core moral beliefs in the course of their work and during the COVID-19 pandemic [6,12,13]. For example, a meta-analysis of 57 studies conducted in 17 countries estimated that 24% (95% confidence interval [CI]:20–28%) of HCWs overall and 25% (95% CI:18–33%) of nurses were at high risk for depressive symptoms [7]. Moreover, a systematic review of suicidal ideation among HCWs during the COVID-19 pandemic reported suicidal ideation in 2.4–21.7% of HCWs [6]. HCWs have been reported to be at a higher risk for depressive symptoms than the general population and those employed in other sectors [4,5]. Furthermore, PCR (Polymerase Chain Reaction)-positive testing has been shown to be related to higher depressive symptoms than PCR-negative testing [14,15], and fear of COVID-19 and workplace stressors were significantly associated with turnover intentions [16,17].

Japan’s healthcare delivery system is predicated upon a universal health insurance regime and an unencumbered accessibility structure that confers patients with the liberty to obtain medical attention at any health institution of their choosing [18]. Furthermore, more than 70% of Japan’s medical institutions are privately owned, and private medical institutions account for more than half of the total beds in the country [19]. Previous studies in Japan have revealed mental health issues among HCWs responding to patients with COVID-19 in hospitals. Many reports have conveyed severe distress or complaints among HCWs in communities with a significant number of COVID-19 cases [20,21,22,23]. For example, a study of HCWs at the Japanese Red Cross Hospital reported that being a nurse and having high anxiety symptoms were risk factors for depression. Only highly resilient HCWs were less likely to develop depression than others [20]. In addition, a study of HCWs at a university hospital showed that young new HCWs developed the highest rate of depression, regardless of occupation [22].

While confusion spread among hospitals and other care facilities in Japan after major COVID-19 outbreaks, Fukushima Prefecture established a crisis management system, providing infection control, arrangement of hospital beds, and mental health support. As part of this multilayered system, the psychosocial support team of the Fukushima Medical University (F-PSST) was launched in 2020, providing psychosocial support for HCWs working at various medical facilities facing severe panic after major COVID-19 outbreaks [24,25,26]. However, few studies have addressed mental health problems among HCWs working at hospitals during unexpected major COVID-19 outbreaks.

Previous studies in Japan have focused on HCWs working at hospitals that could provide highly specialized and intensive care during the COVID-19 pandemic [20,21,22,23]. HCWs in these hospitals were presumably well-prepared and trained to treat patients with COVID-19. In contrast, HCWs in typical hospitals and nursing homes, such as those where the F-PSST provided psychosocial support, were less likely to have received specialized training for COVID-19 care and might be ill-prepared for major COVID-19 outbreaks. In addition, if a major outbreak occurred unexpectedly, their workload would increase substantially due to their being infected with COVID-19 or to a lack of HCWs. Therefore, it would be expected that the mental health of HCWs would be very poor if they experienced a major outbreak in a typical hospital or nursing home, although this has rarely been investigated in previous studies.

This study addressed the mental health issues experienced by HCWs working in ordinary clinical settings during unexpected, major COVID-19 outbreaks, considering the psychosocial support they received. It examined their depressive symptoms retrospectively, using secondary data obtained during psychosocial support performed by the F-PPST. We contend that focusing on HCWs working at various facilities after major COVID-19 outbreaks may shed light on the crucial role of psychosocial support.

## 2. Method

### 2.1. Psychosocial Support System by the F-PSST

As mentioned above, the COVID-19 task force in the Fukushima prefectural office requested that the F-PSST provide psychosocial support for HCWs working at hospitals and nursing homes experiencing major in-house COVID-19 outbreaks [24,25,26]. The F-PSST, which consisted of three clinical psychologists and one psychiatrist, closely cooperated with the infection control and the disaster medical assistance teams. To support these facilities, the F-PSST has provided psychosocial support for HCWs since December 2020. First, the F-PSST offered group psychoeducation for HCWs to share knowledge about common mental health problems encountered after a major COVID-19 outbreak, such as depressive symptoms, sleep difficulty, posttraumatic stress symptoms, and self-management skills, including help-seeking. Second, a self-administered screening test was employed to identify HCWs at risk for mental health issues, especially depressive symptoms and suicidal ideation. Third, psychosocial interventions (mainly individual counseling) with face-to-face, telephone, or video conferencing systems such as Zoom (San Jose, CA, USA: Zoom Video Communications Inc.), based on Mental Health First Aid [27] and the Guide to Mental Health and Psychosocial Support during the COVID-19 pandemic [28,29], were provided by clinical psychologists based on the results of the above tests. These interventions were also provided to HCWs who tested PCR-positive after and during quarantine. While clinical psychologists typically conducted single or two-three sessions for HCWs at high risk of mental health problems, especially depressive symptoms, a psychiatrist dealt with more severe issues, such as suicidal ideation, and sometimes referred HCWs to psychiatric clinics as needed.

### 2.2. Study Design and Participants

This study was conducted with a retrospective multi-institutional research design, using opt-out recruitment methods. It used secondary data from self-administered screening tests and psychosocial support for HCWs working in hospitals and nursing homes where major COVID-19 outbreaks occurred in Fukushima Prefecture, Japan. We operationally defined a major COVID-19 outbreak as 20 or more HCWs and patients/facility users infected with COVID-19 in a hospital or nursing home. The screening test data were not initially collected for research purposes but to provide psychosocial support for HCWs after major COVID-19 outbreaks in the facility. In addition, because major COVID-19 outbreaks in hospitals and nursing homes were incidental events, conducting a prospective study was challenging and impractical. The Institutional Review Board of Fukushima Medical University approved the study protocol (General 2022-078).

Of the 16 hospitals and nursing homes where the F-PSST provided psychosocial support after major COVID-19 outbreaks in Fukushima Prefecture from December 2020 to April 2022, the study included 558 HCWs who worked in eight facilities (five hospitals and three nursing homes) where the screening test was conducted. These eight facilities had approximately 100 to 1000 beds and were typical hospitals and nursing homes that did not provide highly specialized and intensive care for COVID-19.

### 2.3. Measures

#### 2.3.1. Sociodemographic Characteristics

The screening test collected sociodemographic characteristics, such as the facility type, number of days between the outbreak occurrence and response to the screening test, sex, age, job type, and COVID-19 infection status.

#### 2.3.2. Patient Health Questionnaire-9 (PHQ-9)

Depressive symptoms were measured using PHQ-9 [30]. We used the Japanese version of PHQ-9 [31,32]. The PHQ-9 consists of nine items rated on a four-point Likert-type scale ranging from zero (not at all) to three (almost every day) over two weeks. The PHQ-9 has cutoff scores, and the sum scores of zero to four on the PHQ-9 are generally evaluated as zero, 5 to 9 as mild, 10 to 14 as moderate, 15 to 19 as moderately severe, and 20 to 27 as severe [32,33]. In addition, PHQ-9 has shown adequate sensitivity and specificity for assessing depression [31,32]. In the statistical analysis, participants were divided into two groups using a cut-off value of 9 out of 10 for the total score (the probable depression group and others) and 0 out of 1 for the item nine score (the suicidal ideation group and others).

#### 2.3.3. Self-Confidence in Continuing to Work

We measured how a major COVID-19 outbreak affected self-confidence in continuing to work with HCWs. We asked, “Are you confident about continuing to work?” Participants rated their answers on a four-point Likert-type scale ranging from zero (confident) to three (not confident).

#### 2.3.4. COVID-19-Related Symptoms

To identify the influence of COVID-19-related symptoms on mental health among HCWs, we inquired about the symptoms of HCWs relating to sick leave due to COVID-19 soon before or during our screening tests. HCWs were asked to complete a list of 23 COVID-19-related symptoms (e.g., impaired smell, fatigue, and cough).

### 2.4. Statistical Analysis

We determined the measurement items of the screening test based on the size of the COVID-19 outbreak and the opinions of managers and HCWs at each facility. Hence, the facilities where the screening tests were conducted did not necessarily use the same measurement items, and the amount of data for each measurement item differed. For example, self-confidence in continuing to work and COVID-19-related symptoms were only measured in some facilities. We conducted statistical analyses after pairwise deletion of unmeasured items or missing values to include as much available data as possible. We performed all analyses using IBM SPSS Statistics for Windows version 26 (IBM Corp., Armonk, NY, USA). We set statistical significance at *p* < 0.05.

The primary outcomes of this study are data on depressive symptoms. When performing logistic regression analysis, the PHQ-9 total score is converted into dichotomous values, one less than 10 points and the other greater than 10 points. The secondary outcome of this study is suicidal ideation. When conducting logistic regression analysis, item nine score of PHQ-9 is also converted into dichotomous values of 0 and greater than 1 point.

First, to examine differences in the participants’ characteristics for each category, we calculated the descriptive statistics stratified by whole sample and COVID-19 infection status; next, we compared each variable’s frequency or mean score by COVID-19 infection status using Pearson’s chi-squared test for frequency or one-way analysis of variance for a mean score. Second, to examine the association between each sociodemographic variable and depressive symptoms without considering the influence of other variables, we conducted crude logistic regression analyses with each sociodemographic variable as independent variables and the total and item nine scores of PHQ-9 as dependent variables. Third, to investigate the association between each sociodemographic variable and depressive symptoms after controlling for the effects of other variables, we conducted multiple logistic regression analyses with sociodemographic variables as independent variables and the total and item nine scores of PHQ-9 as dependent variables. Fourth, to assess the relationship between depressive symptoms, suicidal ideation, and self-confidence in continuing to work, we conducted partial correlation analyses between the total and item nine scores of PHQ-9 and self-confidence in continuing to work using sociodemographic variables as covariates. Fifth, we calculated the descriptive statistics of COVID-19-related symptoms for PCR-positive HCWs. Finally, to examine factors related to depressive symptoms and suicidal ideation in PCR-positive HCWs after controlling for the influence of other variables, we conducted multiple logistic regression analyses in PCR-positive HCWs; sociodemographic variables and the number of COVID-19-related symptoms were independent variables and the total and item nine scores of PHQ-9 were the dependent variables.

## 3. Results

### 3.1. Sociodemographic Information of Participants

Table 1 presents the sociodemographic information stratified by COVID-19 infection status. The mean ± SD for the number of days between the outbreak occurrence and response to the screening test is 27.28 ± 19.01, and the median is 16.50. There was a large discrepancy between the mean and median because one of the eight facilities was late in requesting support from the COVID-19 task force in the Fukushima prefectural office; this also delayed the implementation of screening tests. Therefore, the median may be more representative of the participants. Among the 558 HCWs, 386 (69.2%) are PCR-negative, 108 (19.4%) are PCR-positive, and 64 (11.5%) had unknown PCR status. Participants comprised 403 (74.4%) women and 139 (25.6%) men. One hundred and thirty-two (24.8%) were aged 20–29 years, 144 (27.1%) were 30–39 years, 117 (22.0%) were 40–49 years, 103 (19.4%) were 50–59 years, and 36 (6.8%) were 60 years or older. The severity of depressive symptoms among all participants was moderate (total PHQ-9 scores = 10–14) in 15.2%, moderately severe (15–19) in 9.5%, severe (20–27) in 4.7%, and moderate or above (≥10) in 29.4% of participants. In addition, 57 HCWs (10.2%) reported suicidal ideation (item nine scores of PHQ-9 ≥ 1).

The PCR-negative group included significantly more hospitals than nursing homes (80.4% vs. 36.8%) and PCR-positive and unknown PCR status groups included significantly fewer hospitals than nursing homes (15.5% vs. 30.1%, 4.1% vs. 32.6%, respectively; *χ*^2^ (2) = 117.81, *p* < 0.001). The PCR-negative group included significantly more other HCWs than nursing home staff (78.5% vs. 39.0%); the PCR-positive group included significantly fewer nurses than other HCWs (12.8% vs. 33.8%); the unknown PCR status group included fewer nursing home staff than other HCWs (8.7% vs. 27.3%; *χ*^2^ (4) = 55.00, *p* < 0.001). The unknown PCR status group included less than 1 for the item nine scores of PHQ-9 than 0 (3.5% vs. 12.4%; *χ*^2^ (2) = 6.00, *p* = 0.05). The unknown PCR status group had significantly higher self-confidence in continuing work than PCR-negative and PCR-positive (*F* (2, 148) = 4.21, *p* = 0.041). No significant differences were observed in the number of days between the outbreak and responses to the screening test, sex, age, and the mean, degree of severity, and ≥10 of PHQ-9 scores.

**Table 1 ijerph-20-04718-t001:** Participants’ sociodemographic characteristics.

		Overall		PCR-Negative		PCR-Positive		Unknown PCR Status		*p* ^a^
		*n*/Score	(%)	*n*/Score	(%)	*n*/Score	(%)	*n*/Score	(%)	
Total ^a^		558	(100.0)	386	(69.2)	108	(19.4)	64	(11.5)	
Facility type										<0.001
Hospital		414	(74.2)	333	(86.3)	64	(59.3)	17	(26.6)	
Nursing home		144	(25.8)	53	(13.7)	44	(40.7)	47	(73.4)	
Number of days between outbreak and response to the screening test										0.076
	Mean	27.28	-	27.4	-	29.27	-	22.95	-	
	Median	16.50	-	15.0	-	20.50	-	25.00	-	
	SD	19.01	-	19.9	-	19.11	-	10.95	-	
	Range	7–88	-	7–71	-	12–88	-	9–64	-	
Sex										0.218
	Woman	403	(74.4)	292	(75.8)	76	(73.8)	35	(64.8)	
	Man	139	(25.6)	93	(24.2)	27	(26.2)	19	(35.2)	
Age (years)										0.064
	20–29	132	(24.8)	107	(28.0)	18	(17.5)	7	(14.9)	
	30–39	144	(27.1)	100	(26.2)	32	(31.1)	12	(25.5)	
	40–49	117	(22.0)	86	(22.5)	21	(20.4)	10	(21.3)	
	50–59	103	(19.4)	66	(17.3)	26	(25.2)	11	(23.4)	
	≥60	36	(6.8)	23	(6.0)	6	(5.8)	7	(14.9)	
Job type										<0.001
	Nurse	218	(40.6)	160	(42.1)	31	(30.1)	21	(38.9)	
	Nursing home staff	77	(14.3)	30	(7.9)	46	(44.7)	12	(22.2)	
	Other HCW	242	(45.1)	190	(50.0)	26	(25.2)	21	(38.9)	
PHQ-9										
	Mean	7.63	-	7.34	-	8.69	-	7.64	-	0.095
	SD	5.75	-	5.59	-	6.20	-	5.79	-	
	Degree of severity									0.442
	0–4 (minimal)	186	(33.3)	130	(33.7)	36	(33.3)	20	(31.3)	
	5–9 (mild)	208	(37.3)	151	(39.1)	32	(29.6)	25	(39.1)	
	10–14 (moderate)	85	(15.2)	57	(14.8)	20	(18.5)	8	(12.5)	
	15–19 (moderately severe)	53	(9.5)	32	(8.3)	12	(11.1)	9	(14.1)	
	20–27 (severe)	26	(4.7)	16	(4.1)	8	(7.4)	2	(3.1)	
	≥10 on total scores (probable depression)	164	(29.4)	105	(27.2)	40	(37.0)	19	(29.7)	0.140
	≥1 on item nine scores (suicidal ideation)	57	(10.2)	39	(10.1)	16	(14.8)	2	(3.1)	0.050
Self-confidence in continuing to work										0.041
	Mean	1.69	-	3.55	-	3.00	-	3.90	-	
	SD	1.24	-	1.02	-	0.91	-	0.74	-	

Note. HCW, healthcare worker; PCR, Polymerase Chain Reaction; PHQ-9, Patient Health Questionnaire-9. ^a^
*p* for Pearson’s chi-square test or one-way analysis of variance.

### 3.2. Associations between Sociodemographic Variables and Depressive Symptoms

Table 2 presents the results of crude logistic regression analyses with each sociodemographic variable as the independent variable and the total and item nine scores of PHQ-9 as the dependent variables. The results show that, among the sociodemographic characteristics, nurses and nursing home staff are significantly associated with higher total and item nine scores of PHQ-9 compared to other HCWs. In addition, PCR-positive HCWs are significantly associated with higher total PHQ-9 scores compared to PCR-negative.

Table 3 presents the results of multiple logistic regression analyses with all sociodemographic variables as independent variables and the total and item nine scores of PHQ-9 as the dependent variable. The results show that being a nurse is significantly associated with higher total and item nine scores of PHQ-9 compared to other HCWs. In addition, ages 30–39 and 50–59 years are significantly associated with lower item nine scores of PHQ-9 compared to ages 20–29 years. 

As Appendix A, the results of multiple logistic regression analyses for the total and item nine scores of PHQ-9 stratified by the period of in-house COVID-19 outbreak occurrence, facility type, sex, age, job type, and COVID-19 infection status are shown in Appendix A.

**Table 2 ijerph-20-04718-t002:** Crude logistic regression analysis for PHQ-9.

		Total Score (≥10 Compared with <10)				Item Nine Score (≥1 Compared with <1)			
		OR	95% CI		*p*	OR	95% CI		*p*
			LL	UL			LL	UL	
Facility type									
	Hospital	Ref				Ref			
	Nursing home	1.08	0.71	1.63	0.722	1.25	0.69	2.29	0.465
Number of days between outbreak occurrence and response to the screening test		0.99	0.98	1.00	0.084	0.98	0.96	1.00	0.079
Sex									
	Woman	Ref				Ref			
	Man	0.70	0.45	1.08	0.106	0.94	0.50	1.77	0.843
Age (years)									
	20–29	Ref				Ref			
	30–39	0.63	0.36	1.09	0.096	0.46	0.21	1.01	0.052
	40–49	1.44	0.85	2.44	0.179	0.89	0.44	1.80	0.741
	50–59	1.23	0.71	2.13	0.468	0.41	0.17	1.01	0.052
	≥60	1.05	0.47	2.34	0.906	0.51	0.14	1.82	0.299
Job type									
	Other HCW	Ref				Ref			
	Nurse	2.74	1.80	4.15	<0.001	2.41	1.26	4.62	0.008
	Nursing home staff	2.06	1.17	3.64	0.013	2.79	1.25	6.27	0.013
COVID-19 infection status									
	PCR-negative	Ref				Ref			
	PCR-positive	1.57	1.00	2.47	0.048	1.55	0.83	2.89	0.171
	Unknown PCR status	1.13	0.63	2.02	0.680	0.29	0.07	1.22	0.091

Note. HCW, healthcare worker; PCR, Polymerase Chain Reaction; PHQ-9, Patient Health Questionnaire-9.

**Table 3 ijerph-20-04718-t003:** Multiple logistic regression analysis for PHQ-9 (*n* = 526).

		Total Score (≥10 Compared with <10)					Item Nine Score (≥1 Compared with <1)				
		OR	95% CI		VIF	*p*	OR	95% CI		VIF	*p*
			LL	UL				LL	UL		
Facility type											
	Hospital	Ref					Ref				
	Nursing home	1.07	0.60	1.91	1.69	0.823	1.53	0.55	4.20	1.69	0.413
Number of days between outbreak occurrence and response to the screening test		0.99	0.98	1.00	1.12	0.101	0.98	0.96	1.00	1.12	0.076
Sex											
	Woman	Ref					Ref				
	Man	1.11	0.68	1.83	1.16	0.667	1.35	0.66	2.76	1.16	0.414
Age (years)											
	20–29	Ref					Ref				
	30–39	0.64	0.36	1.14	1.58	0.132	0.41	0.18	0.93	1.58	0.034
	40–49	1.38	0.79	2.41	1.50	0.256	0.78	0.36	1.69	1.50	0.535
	50–59	1.05	0.59	1.89	1.58	0.863	0.27	0.10	0.75	1.58	0.011
	≥60	1.03	0.45	2.35	1.22	0.947	0.45	0.11	1.89	1.22	0.274
Job type											
	Other HCW	Ref					Ref				
	Nurse	2.61	1.67	4.09	1.26	<0.001	2.61	1.28	5.29	1.26	0.008
	Nursing home staff	1.65	0.84	3.26	1.52	0.148	2.71	0.83	8.88	1.52	0.099
COVID-19 infection status											
	PCR-negative	Ref					Ref				
	PCR-positive	1.57	0.95	2.60	1.15	0.081	1.46	0.73	2.95	1.15	0.286
	Unknown PCR status	1.35	0.66	2.77	1.21	0.405	0.28	0.06	1.38	1.21	0.118
	*R^2^*	0.10	**				0.18	**			

Note. HCW, healthcare worker; PCR, Polymerase Chain Reaction; PHQ-9, Patient Health Questionnaire-9. ** *p* < 0.01.

### 3.3. Association between Depressive Symptoms and Self-Confidence in Continuing to Work

Because this study analyzed the results of a screening test at a single point in time, it is possible that the in-house major COVID-19 outbreak exacerbated depressive symptoms, thereby decreasing self-confidence in continuing work, or that the outbreak exacerbated depressive symptoms, thereby decreasing self-confidence in continuing work. Hence, the causal relationship between depressive symptoms and self-confidence in continuing work remained unclear. In addition, because self-confidence in continuing work was measured only at some facilities, the number of valid responses was small (*n* = 151, 21.7% of all participants). Therefore, we conducted a partial correlation analysis of the total and item nine scores of PHQ-9 and self-confidence in continuing work, controlling for facility type, the number of days between the outbreak and response to the screening test, sex, age, job type, and COVID-19 infection status. The partial rank order correlation coefficient between the total PHQ-9 scores and self-confidence in continuing work is −0.48 (*p* < 0.001), and between the item nine scores of PHQ-9 and self-confidence in continuing to work is −0.44 (*p* < 0.001).

### 3.4. Sociodemographic Information of COVID-19-Related Symptoms in PCR-Positive HCWs

Table 4 presents the sociodemographic characteristics of COVID-19-related symptoms among PCR-positive HCWs. Of the 108 PCR-positive HCWs, 79 (73.1%) were asked about their COVID-19-related symptoms in the screening test, and 68 (86.1%) exhibited at least one COVID-19-related symptom. The most frequently complained COVID-19-related symptoms were decreased physical strength, cough, weakened muscles, fatigue, impaired smell, and sleep problems.

### 3.5. Associations between Sociodemographic Variables, Number of COVID-19-Related Symptoms, and Depressive Symptoms

Table 5 presents the results of multiple logistic regression analyses with all sociodemographic variables; the number of COVID-19-related symptoms were independent variables and the total and item nine scores of PHQ-9 was the dependent variable for 79 PCR-positive HCWs. In all results, the variance inflation factor (VIF) was less than 10, indicating no multicollinearity. The results demonstrate that being a nurse is significantly associated with higher total scores compared to other HCWs, and the number of COVID-19-related symptoms is significantly associated with higher total scores. Furthermore, multiple logistic regression analysis shows that the number of COVID-19-related symptoms is significantly associated with higher item nine scores.

### 3.6. Psychosocial Interventions for At-Risk HCWs

After performing a self-administered screening test, psychosocial interventions were performed from December 2020 to April 2022 for 332 at-risk HCWs (128 with face-to-face intervention and 204 remotely) by the F-PSST. Individual counseling of 30–50 min per session was offered to participants. During these sessions, at-risk HCWs often expressed psychological distress related to job stress during major COVID-19 outbreaks. For example, “With many staff members absent due to infection, my workload has increased, and it has become difficult to sleep,” “As a manager, I have to control the outbreak, but I cannot do so and I feel hopeless,” and “I am depressed because my infection has inconvenienced patients and their families.” Clinical psychologists and psychiatrists attentively listened to and assessed psychological distress from at-risk HCWs, discussing ways to adjust work and deal with issues in their lives.

## 4. Discussion

This study examined depressive symptoms among HCWs working at typical hospitals and nursing homes experiencing in-house major COVID-19 outbreaks. To this end, it used secondary data obtained from psychosocial support sessions performed by the F-PPST. 

This study’s results showed that, first, 29.4% of the study participants exhibited moderate or more severe depressive symptoms, and 10.2% had suicidal ideation. A previous study conducted a systematic review and meta-analysis of depressive symptoms in HCWs during the COVID-19 pandemic [7]. Previously, 16 studies assessed depressive symptoms in HCWs using the same criteria as in the present study (i.e., PHQ-9 scores ≥ 10), showing that those at a higher risk for depression ranged from 8.0% to 44.7%. Only two studies found evidence of higher depression risk than the present study [34,35]. Furthermore, in a systematic review of suicidal ideation among HCWs during the COVID-19 pandemic, 11 studies assessed suicide attempts using the same criteria as in the present study (i.e., item nine scores of PHQ-9 ≥ 1). The proportion of HCWs at high risk of suicidal ideation ranged from 2.4% to 18.6% [6]. These results indicate psychological impacts and burdens among HCWs unexpectedly exposed to major COVID-19 outbreaks at workplace. During the psychological intervention conducted by the F-PSST, many HCWs expressed intense guilt based on moral injury, represented as concern over the failure of protection procedures, insufficient contact with patients or facility users during the pandemic, and feelings of helplessness [12,13] rather than fear of infection. These negative feelings seemed to lower self-esteem and self-confidence, leading to depressive symptoms and sometimes even suicidal ideation in professionals. A previous study revealed a high level of suicidal ideation among HCWs during the COVID-19 pandemic [6]. Hence, special attention should be devoted to HCWs’ suicide risk.

Second, the results of the crude and multiple logistic regression analyses indicated that features such as being a nurse, nursing home staff, and PCR-positive were associated with higher depressive symptoms. Nurses were also associated with higher depressive symptoms when controlling for covariates. These results were consistent with previous studies [36,37,38]. However, multiple logistic regression analysis controlling for the influence of various sociodemographic variables showed that only being a nurse was associated with higher depressive symptoms. In contrast, the association with depressive symptoms disappeared for among nursing home staff and PCR-positive testing. Among HCWs, nurses could be at high risk of mental disorders due to their primary duties, including direct care for patients at high risk of COVID-19 [37]. Meanwhile, as PCR-positive HCWs responded to the screening test soon after and during the quarantine, the association between testing PCR-positive and depressive symptoms might have disappeared after being relieved from hard work. The COVID-19 quarantine might have also reduced the psychological burden and excessive workload experienced by HCWs related to the care of patients with COVID-19.

Third, multiple logistic regression analysis for depressive symptoms showed no association with each age group, while multiple logistic regression analysis for suicidal ideation showed a positive association with being aged 20–29 years. Several previous studies have reported that younger HCWs were a risk factor for suicide ideation; the results of this study were consistent with these [39,40]. HCWs aged 20–29 years may not have adequate professional skills or preparedness for emergencies due to their limited experience in professional practice. Therefore, when a major outbreak unexpectedly occurs in a hospital or nursing home, younger HCWs may experience suicidal ideation owing to feelings of failure or inadequateness and guilt for not responding professionally to the emergency [12].

Fourth, partial correlation analysis results indicated that more depressive symptoms and suicidal ideation were associated with lower self-confidence in continuing to work. When a major COVID-19 outbreak occurred in a typical hospital or nursing home, assessing HCWs for depressive symptoms and providing psychosocial intervention was crucial. Preventing HCWs from leaving their jobs was also vital from a public health perspective [16,17]. Nurses’ high self-confidence in their work and effective engagement with patients with COVID-19 were found to be positively correlated [41]. This study’s results suggest that psychosocial intervention for HCWs in typical hospitals and nursing homes where major COVID-19 outbreaks have occurred may not only improve depressive symptoms and suicidal ideation, but also restore self-confidence in continuing to work and prevent leaving a job.

Fifth, more than 85% of PCR-positive HCWs exhibited COVID-19-related symptoms, and many reported approximately two or more symptoms after returning to work. The most frequently experienced symptoms were decreased physical strength, cough, weakened muscles, fatigue, impaired smell, and sleep problems. Previous studies have reported that these could appear as acute or post-acute COVID-19-related symptoms [14,42,43,44,45]. The psychological intervention conducted by the F-PSST also revealed that many HCWs were willing to continue working while hiding and enduring residual symptoms. In addition, among PCR-positive HCWs, being nurses and the number of COVID-19-related symptoms were associated with more depressive symptoms, and the number of COVID-19-related symptoms were associated with suicidal ideation. Although previous studies reported that PCR-positive testing was associated with higher depressive symptoms and suicidal ideation [14,46], we did not find a significant association between PCR-positive testing and higher depressive symptoms and suicidal ideation controlling for other sociodemographic variables. These results suggest that experiencing multiple COVID-19-related symptoms is more strongly associated with depressive symptoms and suicidal ideation compared to testing PCR-positive.

We provided evidence of intensive psychological reactions among HCWs experiencing unexpected major COVID-19 outbreaks, suggesting that the pre-existing intra-hospital care system may have experienced substantial distress [22,23,27]. Based on a high-risk approach, we performed multi-step interventions as an outside support team (F-PSST), from psychoeducation to psychosocial intervention. Some HCWs were strongly recommended to leave work and receive psychiatric treatment, especially when they exhibited severe depressive symptoms, such as suicidal ideation. In many facilities, HCWs did not receive adequate suggestions or instructions to improve their mental health status, because supervisors in charge of their health management were confused and exhausted during major COVID-19 outbreaks. Therefore, outside crisis responses to in-house major COVID-19 outbreaks, such as the F-PSST, could help organizations and individuals recover as soon as possible.

Despite its contributions, this study has several limitations. First, it was conducted using opt-out recruitment methods, as a retrospective study with informed consent was unfeasible. This study used secondary data obtained during urgent interventions. Therefore, future hypothesis testing studies should examine the validity of its results. Second, the screening test was administered at one-time point before psychosocial interventions; therefore, we could not investigate the course of depressive symptoms after the major COVID-19 outbreak. In this study, 29.4% of HCWs exhibited moderate or higher depressive symptoms, and 10.2% had suicidal ideation. However, higher depressive symptoms and suicidal ideation might be temporary, because the screening tests were conducted during the facility’s early phase of a major COVID-19 outbreak. Therefore, a screening test should be performed at least twice in the future to assess changes in depressive symptoms and suicidal ideation over time. Third, the sample representativeness of HCWs who underwent the screening test was limited. Among the HCWs in typical hospitals and nursing homes that received requests for assistance from the Fukushima Prefecture, the screening tests were only conducted on a limited population. The study’s sample was determined based on the size of the outbreak and the opinions of managers and HCWs in the facility. Therefore, future studies should address HCWs working in hospitals and nursing homes outside the Fukushima Prefecture.

## 5. Conclusions

This study examined depressive symptoms among HCWs who experienced unexpected major COVID-19 outbreaks within typical hospitals and nursing homes. Previous studies have focused on the mental health of HCWs in hospitals that provide specialized treatment for COVID-19. Facilities where major COVID-19 outbreaks unexpectedly occurred included typical hospitals and nursing homes that could not provide specialized treatment for COVID-19 owing to equipment and personnel limitations. HCWs in these hospitals and nursing homes were less likely to receive specialized training in COVID-19 care and might be ill-prepared for major COVID-19 outbreaks. Therefore, this study clarified the mental health characteristics and associated factors of HCWs in these facilities. We showed that major outbreaks of emerging infectious diseases might severely affect the mental health of HCWs. The findings of this study expand the current understanding of HCWs’ mental health status, suggesting the importance of psychosocial support during unexpected major outbreaks in typical health facilities.

## Figures and Tables

**Table 4 ijerph-20-04718-t004:** Sociodemographic characteristics for COVID-19-related symptoms in PCR-positive HCWs (*n* = 79).

COVID-19-Related Symptoms	*n*/Scores	(%)
One or more COVID-19-related symptoms	68	(86.1)
Impaired taste	14	(17.7)
Impaired smell	20	(25.3)
Fatigue	25	(31.6)
Slight fever	8	(10.1)
Dyspnea	5	(6.3)
Headache	18	(22.8)
Chest pain	4	(5.1)
Sore throat	13	(16.5)
Difficulty concentrating	13	(16.5)
Memory impairment	2	(2.5)
Sleep problems	20	(25.3)
Decreased appetite	8	(10.1)
Diarrhea	5	(6.3)
Hair loss	4	(5.1)
Red eyes	0	(0.0)
Dizziness	5	(6.3)
Cough	36	(45.6)
Nasal discomfort	18	(22.8)
Hoarse voice	13	(16.5)
Numbness of the tongue and lips	1	(1.3)
Decreased physical strength	44	(55.7)
Weakened muscle	31	(39.2)
Others	7	(8.9)
Mean	2.91	-
Median	2.00	-
SD	3.37	-
Range	0–19	-

**Table 5 ijerph-20-04718-t005:** Multiple logistic regression analysis for PHQ-9 in PCR-positive HCWs (*n* = 79).

		Total Score (≥10 Compared with <10)					Item Nine Score (≥1 Compared with <1)				
		OR	95% CI		VIF	*p*	OR	95% CI		VIF	*p*
			LL	UL				LL	UL		
Facility type											
	Hospital	Ref					Ref				
	Nursing home	1.34	0.30	6.03	1.80	0.702	1.32	0.15	11.88	1.80	0.806
Number of days between outbreak occurrence and response to the screening test		1.00	0.98	1.03	1.30	0.901	0.97	0.92	1.02	1.30	0.184
Sex											
	Woman	Ref					Ref				
	Man	2.57	0.61	10.73	1.53	0.196	1.27	0.63	2.56	1.15	0.511
Age (years)											
	20–29	Ref					Ref				
	30–39	0.72	0.16	3.29	2.06	0.669	0.17	0.02	1.57	2.06	0.118
	40–49	1.65	0.38	7.23	1.77	0.504	1.80	0.40	8.12	1.77	0.442
	50–59	2.70	0.63	11.58	1.93	0.181	0.23	0.04	1.45	1.93	0.118
	≥60	2.58	0.26	25.52	1.38	0.418	0.91	0.05	17.39	1.38	0.951
Job type											
	Other HCW	Ref					Ref				
	Nurse	4.41	1.01	19.25	2.04	0.049	4.96	0.85	28.85	2.04	0.075
	Nursing home staff	2.06	0.41	10.28	2.02	0.380	4.63	0.64	33.24	2.02	0.128
Number of COVID-19-related symptoms		1.45	1.16	1.81	1.09	0.001	1.22	1.01	1.47	1.09	0.042
	*R^2^*	0.39	**				0.38	*			

Note. ** *p* < 0.01; * *p* < 0.05.

## Data Availability

Data is unavailable due to privacy or ethical restrictions.

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
