# Peer review of "Impact of Unexpected In-House Major COVID-19 Outbreaks on Depressive Symptoms among Healthcare Workers: A Retrospective Multi-Institutional Study"

_ijerph, 2023, doi:10.3390/ijerph20064718_

Round 1

Reviewer 1 Report

Dear Authors,

Over all, the study focused an important issues among HCWs during the period of COVID19 outbreak. The article has been written well and structured. However, in my opinion, the paper has some notable changes regarding methods (Design, measurement tool and analysis ) and text overlap. I have provided numerous remarks and comments and to some extent in the attached PDF file of article.

Author Response

Thank you very much for review of the manuscript (ijerph-2228535) that we send on January, 3rd, 2023. We thank you for providing constructive comments regarding the improvement of the original manuscript. Based on the comments from the three reviewers, major revisions included (1) reanalyzing the PHQ-9 as a dichotomous variable rather than a continuous variable, (2) adding an analysis of the item nine scores of PHQ-9 (i.e., suicidal ideation) as a secondary outcome, and (3) adding subgroup analysis results to the supplementary materials. Please check again as we have made the following corrections to each comment. The revised manuscript was proofread by an expert.

Please see the attachment in detail.

Reviewer 2 Report

The topic of the manuscript is interesting and gives a new insight into the burden on healthcare workers during the Covid 19 pandemic. The introduction is well done with a good presentation of the topic and the specific situation in Japan. However, there are some points left open in the statistical analysis that need to be improved. In particular, it should be mentioned here whether the presumably linear regression of PHQ-9 score values is appropriate or whether modelling PHQ-9 severity groups is more appropriate. In addition, an analysis of the risk factors for suicidal ideation might be interesting. In any case, more details on modelling need to be given and the presentation of results needs to be improved. Also, the study period of December 2020 to April 128 2022 is very wide. I can imagine that there may have been a trend over time in the burden of health workers. This should be presented and discussed.

Below are some more detailed comments on the individual manuscript sections:

Abstract: The abstract should contain more details so that the following questions can be answered by the abstract: How were depressive symptoms assessed? (Please name PHQ-9.) In which time period were the symptoms recorded? Compared to which other professions did the multiple regression analysis show that nurses and nursing home staff are associated with higher depressive symptoms?

Introduction: The introduction is well done. However, I wonder what additional benefit the reference to the cruise ship has. I would not mention this.

Methods:

Lines 170-171: Please use the term "dependent variable" instead of "objective variable".

Was PHQ-4 modelled as a continuous variable or as a 4-level or 5-level variable in terms of severity? An analysis of risk factors for severe or more severe symptoms could also be informative.

Results:

Line 191/192: Instead of the expression "unknown PCR-negative or PCR-positive", it might be more appropriate to speak of an unknown PCR status.

Table 1: P-values are never equal to zero. Please indicate <0.001 in the corresponding entries. Please also provide p-value for comparing PHQ-9 severity. I understand the p-value of 0.042 as the comparison of the mean PHQ-9 score. Is that correct? Please clarify „>=10“ and „≥ 1 on item nine scores“ in Table 1.

The authors should consider whether it would be useful to list the 64 individuals with unknown COVID-19 infection status in an additional column in Table 1.

Lines 214-216: The estimators from Table 2 do not have to be mentioned again in the text. This impedes the flow of reading.

Table 2: Please specify the reference category for discrete influence variables. For example: Do men have lower PHQ-9 scores compared to women? It should also be stated that "20-29", "30-39" etc. characterise the age groups. Please also indicate the reference category at age. What about the HWC >= 60 years? Also make clear that being a nurse or being nursing home staff is always associated with higher PHQ-9 scores compared to other HCW.

Table 3: Why were non-significant parameters from the simple regression analysis included in the multiple model (e.g. "number of days between outbreak and response to screening test")? Please also note the same criticisms of the construction of Table 3 as for Table 2.

Lines 223-227: Why was no regression analysis done on „self-confidence in continuing to work“? I would be more interested in these results than a simple regression coefficient.

Table 4: I don't find this table very interesting. In my opinion, a presentation of these results in the appendix would be sufficient. I would find it more interesting to expand Table 5 to include PCR-negative HCWs, as these can also experience a burden. Please also consider the same points of criticism in the preparation of Table 5 as in Tables 2 and 3.

Wouldn't it also be interesting to know what risk factors are for suicidal ideation? The issue of suicidal ideation has been mentioned several times within the manuscript, but not further statistically evaluated.

Lines 244-256: How was the influence of psychosocial interventions on the PHQ-9 investigated? The results should be presented.

Conclusion: I cannot understand the following conclusion because the study did not examine society but selected HWCs: „We showed that major outbreaks of emerging infectious diseases might severely affect society.“ Please clarify.

Author Response

(The authors gave the same response as above.)

Reviewer 3 Report

In this manuscript, the authors aim to reveal the mental health status of  healthcare workers and show that  high depressive symptoms exists during COVID-19 pandemic, which is a timely topic. However, there exists some major concerns as follows:
1.The author should clarify the contributions more clearly compared with the existing works in recent three years. Also, some differences in term of result or method need to be mentioned to show the advantages of this work.
2.The research question must be stated before result analysis, which is not clear for readers.
3.For the method, only multiple regression analysis is performed. The author should make a deep analysis, i.e., multiple group analysis, to show the difference between groups,e.g., facility type, age, sex of HCW. Besides, the causal relationship between the sociodemographic variable and Symptoms should be further analyzed.
4.After analyzing, the discussion part need to be extended and give deep analysis. Especially, responding to the proposed research questions at beginning. The implication and limitation need to be described.
5.For the questionnaire distribution, the authors further clarify how to recruit the respondents among hospital and nursing home for the long duration from December 2020 to April 2022, e.g., divided into several stages. The variation of Symptoms for HCWs during this period need to be focused.
6. The authors should give some basic information about japans’ hospital which survey in this manuscript, e.g., the scale of hospital, which level among nation medical system. 

Author Response

(The authors gave the same response as above.)

Round 2

Reviewer 1 Report

All the comments have been amended. Thank you

Reviewer 2 Report

Dear Mr. Sato,

Thank you very much for the well revised manuscript. The additional statistical analysis of the 9th item of the PHQ-9 is very informative.

All my questions were answered sufficiently.

However, I have one small comment on Table 3. The abbreviation "VIF" is not further mentioned in the table or in the manuscript and should be explained.

Kind regards

Reviewer 3 Report

No further comment.
